# Feasibility of Dynamic Inhaled Gas MRI-Based Measurements Using Acceleration Combined with the Stretched Exponential Model

**DOI:** 10.3390/diagnostics13030506

**Published:** 2023-01-30

**Authors:** Ramanpreet Sembhi, Tuneesh Ranota, Matthew Fox, Marcus Couch, Tao Li, Iain Ball, Alexei Ouriadov

**Affiliations:** 1Department of Physics and Astronomy, The University of Western Ontario, London, ON N6A 3K7, Canada; 2Faculty of Engineering, School of Biomedical Engineering, The University of Western Ontario, London, ON N6A 3K7, Canada; 3Lawson Health Research Institute, London, ON N6C 2R5, Canada; 4Siemens Healthcare Limited, Montreal, QC H4R 2N9, Canada; 5Department of Chemistry, Lakehead University, Thunder Bay, ON P7B 5E1, Canada; 6Philips Australia and New Zealand, Sydney 2113, Australia

**Keywords:** inert fluorinated gas MRI, fluorine-19, lung magnetic resonance imaging, stretched exponential model, compressed sensing

## Abstract

Dynamic inhaled gas (^3^He/^129^Xe/^19^F) MRI permits the acquisition of regional fractional-ventilation which is useful for detecting gas-trapping in lung-diseases such as lung fibrosis and COPD. Deninger’s approach used for analyzing the wash-out data can be substituted with the stretched-exponential-model (SEM) because signal-intensity is attenuated as a function of wash-out-breath in ^19^F lung imaging. Thirteen normal-rats were studied using ^3^He/^129^Xe and ^19^F MRI and the ventilation measurements were performed using two 3T clinical-scanners. Two Cartesian-sampling-schemes (Fast-Gradient-Recalled-Echo/X-Centric) were used to test the proposed method. The fully sampled dynamic wash-out images were retrospectively under-sampled (acceleration-factors (AF) of 10/14) using a varying-sampling-pattern in the wash-out direction. Mean fractional-ventilation maps using Deninger’s and SEM-based approaches were generated. The mean fractional-ventilation-values generated for the fully sampled k-space case using the Deninger method were not significantly different from other fractional-ventilation-values generated for the non-accelerated/accelerated data using both Deninger and SEM methods (*p* > 0.05 for all cases/gases). We demonstrated the feasibility of the SEM-based approach using retrospective under-sampling, mimicking AF = 10/14 in a small-animal-cohort from the previously reported dynamic-lung studies. A pixel-by-pixel comparison of the Deninger-derived and SEM-derived fractional-ventilation-estimates obtained for AF = 10/14 (≤16% difference) has confirmed that even at AF = 14, the accuracy of the estimates is high enough to consider this method for prospective measurements.

## 1. Introduction

Inhaled gas (^3^He/^129^Xe/^19^F) MRI has been proven to be useful for dynamic lung imaging [1,2,3]. These techniques enable the acquisition of regional fractional-ventilation [4,5,6] measurements which are very useful as CT-alternatives for detecting gas trapping in lung diseases such as lung inflammation, fibrosis, and Chronic Obstructive Pulmonary Disease (COPD) [1]. A potential alternative for using hyperpolarized gases for functional lung MR imaging can be seen with thermally polarized fluorinated gas tracers such as sulfur hexafluoride (SF_6_), hexafluoroethane (C_6_F_6_), and perfluoropropane (PFeP) (C_3_F_8_) [7].

Thus, free-breathing ^19^F dynamic lung imaging has been recently demonstrated in human lungs [7]. This free-breathing wash-out scheme ensures the gradual wash-out of ^19^F gas within the ^19^F MRI lung images obtained from a COPD patient for eight wash-out breaths [7].

Using fluorinated gases provide multiple advantages, such as the ability to be mixed with O_2_ to restore initial magnetization (rather than lose it), shortened imaging times, and increased tolerable breath-holds for patients [8]. The feasibility and effectiveness of fluorine-19 (^19^F) MR imaging of the human lungs has been demonstrated throughout various studies. A study by Pavlova et al. concluded that by using a gas mixture of 80% octafluorocyclobutane (OFCB, C_4_F_8_) and 20% oxygen, they were able to capture ^19^F lung imaging at low magnetic field strengths and at long imaging times which were tolerable due to O_2_ [9]. In addition, a similar study which used perfluorocyclobutone (PFCB, C_4_F_8_) as a visualized fluorinated gas, the authors were able to obtain important information about the lung function through the acquisition of ^19^F pulmonary MRI [10]. This study showed an approach that did not use breath-holding but could still acquire ^19^F-MRI lung images, which is important for patients with COPD or other pulmonary diseases [10]. Shepelytskyi et al. found that lung images acquired using OFCB showed significantly higher normalized SNR compared with PFP, the most common gas agent used in recent preclinical literature [3]. Furthermore, studies have confirmed the feasibility of ^19^F gas MRI using OFCB as a promising inhalable contrast agent, even at lower magnetic field strengths [9]. Gutberlet et al. used free-breathing dynamic ^19^F gas MRI to quantify regional lung ventilation in patients with COPD and concluded that it was feasible at 1.5T [7]. Additionally, Maunder et al. demonstrated the benefits of steady-state free precession (SSFP) for ^19^F C_3_F_8_ gas at 1.5T, as they were able to produce high quality lung ventilation images [11]. Furthermore, a recent study illustrated the use of compressed sensing (CS) to significantly shorten scan acquisition times, which in turn reduces participant breath hold times for ^19^F-MRI of inhaled perfluoropropane [12]. Many recent studies have used various fluorinated gases, thus serving as a backbone towards supporting our techniques and methodology to further investigate the usage of fluorine-19 MRI.

^19^F-gas MRI looks very promising for dynamic lung imaging as it is relatively inexpensive (does not require a polarizer) and can be mixed with O_2_ leading to a shortening of T_1_ [13]. This is the benefit of using this contrast agent in comparison with ^3^He/^129^Xe. Despite this, low SNR due to the thermo-polarized nature of this agent is clearly a limitation. The only way to overcome this limitation is increasing the signal by conducting a signal average. Unfortunately, breath hold time is another obstacle preventing us from an uncountable signal averaging. One solution of this problem is to substantially decrease the acquisition time by using a CS approach, which has some further limitations on the number of under-sampled k-space points. To overcome this CS limitation, the SEM method coupled with CS could be used [14]. SEM utilizes prior knowledge about the signal decay for example diffusion decay [15], relaxation decay, or decay due to the reduction of the contrast agent concentration [16]. Depending on the quantity of captured images in the signal decay curve and this prior knowledge, the acceleration factor (AF) may be increased up to 10 or 14 [17]. 

In this study, our goal was to apply the Stretched Exponential Method (SEM) combined with CS to the dynamic (^3^He/^129^Xe/^19^F) MRI data previously published for normal rats [6,18] and to investigate the influence of acceleration on the accuracy of the SEM-based regional fractional-ventilation estimates. We investigated the potential of accelerated dynamic SEM-based measurements for three different cases: (1) a fully sampled k-space, (2) a 90% retrospectively under-sampled k-space in the wash-in/wash-out direction, (acceleration factor (AF) = 10), and (3) a 93% retrospectively under-sampled (AF = 14) k-space. The sparsity pattern was varied for each k-space in the wash-in/wash-out direction.

In order to generate the SEM-based regional fractional-ventilation maps, we have adapted the SEM equation [8] to fit dynamic wash-in/wash-out data. We hypothesize that the SEM equation can be adapted for fitting the gas density dependence of the MR signal similarly to the fitting time or b-value dependences [14,19].

Finally, we compared the SEM-based fractional-ventilation values with Deninger’s approach based estimates [4], in order to have independent confirmation of the accuracy of the generated fractional-ventilation estimates. 

## 2. Theory

### Stretched Exponential Model (SEM)

Each new wash-out breath of air replaces some volume of the inhaled gas in lung, so the signal intensity of the resulting images was gradually attenuated (Figure 1A).

The following equation can be fitted to the wash-out data when the MR signal does not depend on the flip angle and longitudinal relaxation time, it is also known as a modified Deninger method [4] or FAVOR [5]: *S*(*n*) = *S*_0_(1 − *r^n^*)(1)
where *S*_0_ is the initial signal, *n* is the breath number, *S*(*n*) is the signal intensity after the n-th wash-out breath and *r* is the fractional-ventilation parameter (0 < *r* < 1) [5,18]. *r* can be expressed as the fraction between fresh gas entering the lung and the total volume of gas within the lung (*V_total_*) [5,18]:*r* = *V_new_*/*V_total_ or V_new_*/(*V_new_* + *V_old_*)(2)

The SEM equation not requiring any underlying lung physiology [20] can be used for fitting any signal decays including the gas density dependence of the MR signal (Figure 2):
(3)$$S(n)=S_{0}\exp[-{\left(nr^{\prime }\right)}^{\beta }]$$
where β is heterogeneity index (0 < β < 1), n is the image number, and $r^{\prime }$ is the apparent fractional-ventilation parameter [8]. 

This interpretation allows us to consider the MR signal intensity variation as a reflection of the underlying gas-density variation and, hence, the reconstruction of the under-sampled k-space sets using the adapted SEM equation combined with CS. Thus, the CS-based reconstruction involving all under-sampled k-spaces (not an independent reconstruction) combining with the SEM approach providing the additional information about the signal behavior (a prior knowledge about a system helping overcome the significant under-sampling) can be achieved across all k-spaces [21]. As a result, lung fractional-ventilation maps can be generated using reconstructed images.

The probability density function (*P*) can be used to quantify the Gaussian and non-Gaussian (likely due to the lung disease) distribution of fractional ventilation using the general signal equation [19,20]:(4)$$S(n)=\;S_{0}\int_{0}^{1}P(r)\exp(-r\cdot \bar{n})\;dr$$
where $\bar{n}$ is a *n*-value array or vector, and *S*(*n*) is the signal at a particular *n* (*S*_0_ at *n* = 0 and so on). The inverse Laplace transform of *S*(*n*) can be used to obtain *P*(*r*) for specific analytical representations of the signal attenuation [19].

For the SEM case, the inverse Laplace transformation of Equation (4) yields the probability density function as previously described [19]:(5)$$P(r)=\;\frac{B/r^{\prime }}{{\left(r/r^{\prime }\right)}^{\left(1-\beta /2)/(1-\beta \right)}}\exp\left(-\frac{(1-\beta )\beta^{\beta /(1-\beta )}}{{\left(r/r^{\prime }\right)}^{\beta /(1-\beta )}}\right)f(r);$$
where *f*(*r*) is the auxiliary function:(6)$$f(r)=\;\left\{\begin{array}{l} 1/[1+C{\left(r/r^{\prime }\right)}^{\left(0.5\beta -\beta^{2}/(1-\beta )\right)}],\\ \beta \leq 0.5,\;\\ 1+C{\left(r/r^{\prime }\right)}^{\left(0.5\beta -\beta^{2}/(1-\beta )\right)},\\ \beta >0.5 \end{array}\right\};$$
where parameters *B* and *C* are fitting parameters, and functions of *β* which is a dimensionless quantity [19]. $r^{\prime }$ and *β* maps permit the calculation of *P*(*r*) distributions. The probability density function can be used to generate the SEM-based mean fractional-ventilation parameter. Figure 3 shows the probability density functions plotted for three gases. 

## 3. Methods

### 3.1. Animal Preparation

Thirteen normal rats were used in this study, seven/(five) rats were scanned using ^129^Xe/(^3^He) and six rats were scanned using ^19^F MRI. All animals were used following specific protocols approved by local ethics. Sprague–Dawley rats were used for this study and the method for preparation of rats was followed as described by previous studies [6,18]. The rats were anesthetized through intravenous administration, intubated with a 5-F polypropylene urinary catheter, and ventilated using a custom pneumatic ventilator suitable for MR imaging of hyperpolarized noble gases. The ventilator allowed for the control distribution of tidal volumes [18] and peak inspiratory pressure (PIP) [5]. A detailed description of the custom ventilator system used has been discussed previously [6].

Hyperpolarized ^3^He or ^129^Xe gas was allocated into 300 mL Tedlar plastic bag [5,6] and implanted into a pressured reservoir [6]. The reservoir maintained a constant pressure of 30 cm H_2_O, which allowed inspiratory pressure and tidal volumes of the hyperpolarized gases to be controlled. By administering bags of (^4^He/^129^Xe and O_2_ using a mixture of 80/20) to a representative rat, flow restrictors were used to account for the disparities between the two gas types [6]. PIPs and tidal volumes were calibrated by manometry and water displacement, respectively. 

Breath-holds and tidal breathing (3 mL based on the average size of rats) with air/oxygen or inert fluorinated gas/oxygen mixture were controlled by the ventilator [18]. Imaging of the lungs were obtained in the beginning of the inert fluorinated gas/oxygen mixture (time of breath hold = 10 sec, pressure =12–15 cm H_2_O during breath hold, tidal volume = 8 mL/kg). A washout breathing scheme was used for ^19^F imaging and the protocol incorporated rat lungs saturated with an inert fluorinated gas/oxygen mixture (80/20), for three minutes of continuous breathing at a rate of 60 breaths/min [18]. After the three-minute mark, the fluorinated gas/O_2_ mixture was stopped and a 10 s breath hold was conducted to collect a baseline image. To obtain the second image, we delivered one washout breath of pure O_2_ followed by a 10 s breath hold. This washout breathing technique was repeated nine successive times to ensure complete elimination of fluorinated gas from the rat lungs and to fully sample the washout curve using MR imaging. From the breath hold durations controlled using the ventilator, data acquisition was gathered. At the end of the experiment, all rats were euthanized through intravenous injection of 340 mg/mL of Euthansol in the tail vein (Schering Inc Canada, Point-Claire QC).

### 3.2. MR Imaging

Two gases (^3^He, ^129^Xe) that are normally used for hyperpolarized gas pulmonary MRI were utilized in this animal study as well. Pulmonary MR imaging was performed using a GE 3T MR750 scanner with a high-performance gradient coil (G = 0.5 T/m, slew rate = 2000 T/m/s) and the commercial rat-sized ^3^He (97.3 MHz) and ^129^Xe (35.34 MHz) transmit-receive bird-cage coils (Morris Instruments, Ottawa, Canada) as described previously [6]. Using a spin-exchange optical pumping system, ^3^He gas was polarized with a turnkey Helispin system ensuring 40% polarization after 24 hours of polarization process [6]. Prior to the transfer of the hyperpolarized ^3^He, a Tedlar bag was cleaned three times with medical-grade N_2_ gas and vacuumed (100 mtorr) to reduce ^3^He gas depolarization that could occur by interactions with paramagnetic O_2_. Using a home-build continuous flow polarizer with a gas mixture of 1% Xe, 10% N_2_, and 89% ^4^He, naturally abundant Xe gas (26% ^129^Xe) was polarized to 15%. ^129^Xe was put into a Tedlar bag and thawed after cryogenic separation.

A variable flip angle (VFA) fast gradient-recalled echo method with Cartesian sampling was used to produce 2D projection images. The VFA trajectory was calculated following the FAVOR method [5]. Two-dimensional projection images were obtained according to the parameters: FOV = 40 × 40 mm^2^, matrix = 64 × 64, producing an in-plane resolution of 0.63 mm [6]. Images obtained were whole-lung 2D projections because no slice selection was used. Imaging for ^3^He used TR = 3 ms, TE = 0.6 ms, and bandwidth = 31 kHz, while for ^129^Xe used TR = 14 ms, TE = 2 ms, and bandwidth = 2 kHz [6]. To reduce the T_2_* decay and diffusion-induced signal attenuation caused by imaging gradients, ^3^He imaging was completed with a short echo time. The VFA RF pulse trajectory was calculated for each breath (i.e., image) [5]. Calibration of the RF pulses was achieved through adjustment of the transmitter gain until there was no measurable change in signal over 128 pulses for the entire sample, following a single ^3^He/^129^Xe breath. For the calibration of VFA, five to eight breaths of ^3^He/^129^Xe were required [6].

The ventilator was switched back to air breathing for 2 min after delivering 10 s anoxic breaths [6] to avoid a significant compromising of the animal’s physiology. Two fractional ventilation maps were acquired for each coronal and axial plane for each of the gases. 

All inert fluorinated gas in vivo measurements were performed using a 3.0 T Philips Achieva scanner with maximum gradient strengths of 0.04 T/m. A home-built rat-sized (9 cm inner diameter and 6.8 cm length) quadrature transmit/receive coil tuned to the ^19^F resonance frequency of 120.15 MHz was used for multi breath ^19^F rat lung MR imaging. Two-dimensional whole rat lung projection sulfur hexafloride (SF_6_, rat-1, rat-2, and rat-5) and perfluoropropane (PFP, C_3_F_8_, rat-3, rat-4, and rat-6)) images were obtained in the axial and coronal planes using two-breath acquisitions of 2D X-centric (TE = 0.54 ms, TR = 4 ms for SF_6_, and TR = 20 ms for PFP, 6 × 6 cm^2^, 64 × 64 pixels, Ernst Angle = 70°, BW = 400 Hz/pixel for SF_6_, and BW = 300 Hz/pixel for PFP, 60 averages for SF_6_ and 12 averages for PFP) [18]. Measurements were performed following the breathing scheme as described previously [18]. Because only half of k-space (50.5% of the readout window) was collected in each of the 9 washout-breaths (as well as for baseline), the entire washout protocol was repeated using the opposite readout gradient polarity in order to create a fully sampled k-space data set for reconstruction [18].

### 3.3. Image Processing and Analysis

A Hann filter was applied to all ^19^F k-space data, to maximize the signal-to-noise ratio (SNR) prior to Fourier transformation (IDL 6.4) [16]. ^3^He/^129^Xe data was obtained using wash-in scheme, we reversed the image order to be able to fit the dynamic data with the SEM equation requiring the signal decay not signal growing. A *n* = 0 image was chosen to create a binary mask by using a seeded region-growing algorithm to separate the lungs from the surrounding background and to remove large airways using the custom-built IDL 6.4 algorithm. A binary mask was then applied to the seven remaining ^3^He/^129^Xe wash-in images or eleven remaining ^19^F wash-out images in the series for each animal. 

A fitting algorithm from Abascal et. al. [21] (MATLAB R2020a MathWorks, Natick, MA) was used to fit Equation (3) to the images as a function of *n* and to generate $r^{\prime }$ and *β* maps on a voxel-by-voxel basis. *P*(*r*) distributions were calculated based on Equations (5) and (6) with $r^{\prime }$ and *β* computed on a voxel-by-voxel basis (MATLAB R2020a) using 0.4 and 0.75 as the initial values, correspondingly. The mean fractional ventilation estimates were calculated as the expectation values of the correspondent *P*(*r*).

Two k-space masks mimicking CS-based acceleration were retrospectively applied to the fully sampled ^3^He/^129^Xe/^19^F k-space data (Figure 1A) in order to obtain under-sampled k-space data with the different AFs (Figure 1B,C and Appendix A). Three cases were explored for two different imaging methods (FGRE and X-centric): (1) AF = 1 or no acceleration, (2) AF = 10, 7 k-space lines out of 64 per image using retrospective k-space under-sampling in the imaging direction employing a different under-sampling pattern for each n, and (3) AF = 14, 5 k-space lines out of 64 per image, with retrospective k-space under-sampling as (2). SEM-based k-space reconstruction using the regularization parameters previously determined [22] and regional fractional-ventilation estimates [23] calculation were conducted using Abascal’s algorithm as previously described [21]. The lowest acceleration factor was chosen based on the previous studies, showing a reasonable ratio between the number of the signal decay images and AF [17]. A single binary mask (specific for each gas and rat) for all AFs was used to ensure the same nominal pixel resolution across all the reconstructed images. 

Deninger’s approach was used to calculate the ground truth regional fractional-ventilation estimates using Equation (1), for all the reconstructed images following the SEM-based reconstruction (i.e., all AFs) as previously described [18]. This approach was used to analyze only the reconstructed images including the original images and images obtained after CS-based reconstruction. The hyperpolarized gas images were not corrected for the RF pulse history and T_1_-decay for simplicity (for hp gases the signal level depends on the gas density in lung and leftover magnetization, decaying due to the RF pulses and oxygen induced T_1-_decay) and mimicking a high SNR ^19^F MRI-based data. 

### 3.4. Statistical Analysis

Voxel-by-voxel absolute differences between the regional fractional-ventilation maps generated from the fully sampled and retrospectively under-sampled (AF = 10/AF = 14) data were quantified using: (7)$$Absolute\;Difference\;\;=\sum_{i=1}^{N}\sum_{j=1}^{M}\left|[\frac{FullySampled_{ij}\;-\;UnderSampled_{ij}\;}{FullySampled_{ij}}]\right|\cdot 100\%$$
where *N* and *M* are the corresponding image matrix sizes. 

MANOVA analysis using SPSS Statistics, V22.0 (SPSS Inc., Chicago, IL, USA) was performed to compare the mean regional fractional-ventilation estimates obtained from the fully sampled and retrospectively under-sampled (AF = 10/AF = 14) data. In all statistical analyses, the results were considered significant when the probability of making a Type I error was less than 5% (*p* < 0.05). 

## 4. Results

### Accelerated SEM-Based Dynamic Ventilation

Figure 4, Figure 5 and Figure 6 show representative ^3^He/^129^Xe/^19^F MRI-based fractional ventilation maps generated using the Deninger method (^D^) and the SEM (^S^) from normal animals using two different imaging approaches (FGRE and X-Centric).

The top panel shows fractional ventilation maps calculated for the original fully sampled k-space. The middle and bottom panels show the maps generated for the retrospectively under-sampled data mimicking AF = 10 and 14, correspondingly. The mean values of all ^3^He/^129^Xe/^19^F MRI-based fraction ventilation parameters are summarized in Table 1, Table 2 and Table 3 correspondingly. The spatial distributions of all the fraction ventilation parameters for both imaging methods and the three acceleration factors were relatively homogeneous for all the gases. The mean fractional ventilation values generated for the fully sampled k-space case using the Deninger method were not significantly different from the other fractional ventilation values generated for the non-accelerated/accelerated data using both Deninger and SEM methods (*p* > 0.05 for all cases/gases, except ^3^He, where the mean r values obtained with the SEM and Deninger methods were significantly different (*p* < 0.01)).

For the ^3^He FGRE case the mean absolute differences (Equation (7)) of 5.0%/(6.5%) and 5.0%/(7.0%) were observed between AF = 1 and AF = 10/(AF = 14) for the fractional ventilation values calculated with the Deninger method and the estimates calculated with SEM (Table 1). 

The mean absolute differences of 4.5%/(7.5%) were observed between AF = 1 and AF = 10/(AF = 14) for the fractional ventilation values calculated with the Deninger method for the fully sampled and under-sampled k-space data (Table 1). The mean absolute differences of 4.0%/(6.0%) were observed between AF = 1 and AF = 10/(AF = 14) for the fractional ventilation values calculated with SEM for the fully sampled and under-sampled k-space data (Table 1).

For the ^3^He X-Centric case the mean absolute differences (Equation (7)) of 5.0%/(7.5%) and 5.0%/(8.0%) were observed between AF = 1 and AF = 10/(AF = 14) for the fractional ventilation values calculated with the Deninger method and the estimates calculated with SEM (Table 1). The mean absolute differences of 4.5%/(5.0%) were observed between AF = 1 and AF = 10/(AF = 14) for the fractional ventilation values calculated with the Deninger method for the fully sampled and under-sampled k-space data (Table 1). The mean absolute differences of 4.0%/(4.0%) were observed between AF = 1 and AF = 10/(AF = 14) for the fractional ventilation values calculated with SEM for the fully sampled and under-sampled k-space data (Table 1). Appendix A shows the ^3^He MRI-based fractional ventilation values obtained from all rats with Deninger method and SEM for three acceleration factors and two sampling schemes. 

For the ^129^Xe FGRE case the mean absolute differences (Equation (7)) of 6.5%/(12.0%) and 6.5%/(13.0%) were observed between AF = 1 and AF10/(AF = 14) for the fractional ventilation values obtained with the Deninger method and the estimates calculated with SEM (Table 2). The mean absolute differences of 7.0%/(10.0%) were observed between AF = 1 and AF = 10/(AF = 14) for the fractional ventilation values obtained using the Deninger method for the fully sampled and under-sampled k-space data (Table 2). 

The absolute mean differences of 4.5%/(7.0%) were observed between AF = 1 and AF = 10/(AF = 14) for the fractional ventilation values calculated with SEM for the fully sampled and under-sampled k-space data (Table 2). 

For the ^129^Xe X-Centric case, the mean absolute differences (Equation (7)) of 6.4% (13%) and 6.4%/(13%) were observed between AF = 1 and AF10/(AF = 14) for the fractional ventilation values obtained with the Deninger method and the estimates calculated with SEM (Table 2). The mean absolute differences of 7.5%/(8.0%) were observed between AF = 1 and AF = 10/(AF = 14) for the fractional ventilation values calculated using the Deninger method for the fully sampled and under sampled k-space data (Table 2). The mean absolute differences of 6.0% (5.0%) were observed between AF = 1 and AF = 10/(AF = 14) for the fractional ventilation values calculated with SEM for the fully sampled and under-sampled k-space data (Table 2). 

Appendix A shows the ^129^Xe MRI-based fractional ventilation values obtained from all rats with the Deninger method and SEM for three acceleration factors and two sampling schemes (*p* > 0.5 for all cases). 

For the ^19^F FGRE case the mean absolute differences (Equation (7)) of 15%/(16%) and 15%/(10%) were observed between AF = 1 and AF = 10/(AF = 14) for the fractional ventilation values calculated with the Deninger method and the estimates calculated with SEM (Table 3).

The mean absolute differences of 12.5%/(14%) were observed between AF = 1 and AF = 10/(AF = 14) for the fractional ventilation values calculated with the Deninger method for the fully sampled and under-sampled k-space data (Table 3). The mean absolute differences of 8% (12%) were observed between AF = 1 and AF = 10/(AF = 14) for the fractional ventilation values calculated with SEM for the fully sampled and under-sampled k-space data (Table 3).

For the ^19^F X-Centric case the mean absolute differences (Equation (7)) of 15%/(12.5%) and 15%/(14%) were observed between AF = 1 and AF = 10/(AF = 14) for the fractional ventilation values calculated with the Deninger method and the estimates calculated with SEM (Table 3). The mean absolute differences of 14.0%/(15.0%) were observed between AF = 1 and AF = 10/(AF = 14) for the fractional ventilation values calculated with the Deninger method for the fully sampled and under-sampled k-space data (Table 3). The mean absolute differences of 9%/(9%) were observed between AF = 1 and AF = 10/(AF = 14) for the fractional ventilation values calculated with SEM for the fully sampled and under-sampled k-space data (Table 3).

Appendix A shows the ^19^F MRI-based fractional ventilation values obtained using all the rats with Deninger method and SEM for the three acceleration factors and two sampling schemes (*p* > 0.5 for all cases).

## 5. Discussion

In this work, we studied the combination of CS with an extended stretched-exponential model (SEM) to analyze dynamic ^3^He/^129^Xe/^19^F images in order to accelerate dynamic ventilation in the rat lung and made a number of important findings: (i) for the first time we demonstrated the feasibility of the inhaled gas SEM-based accelerated dynamic ventilation with AF = 10 and 14 in small animals; (ii) SEM-based regional fractional ventilation parameters were found to be similar (not significantly different) to those calculated using Equation (1) or the traditional method; (iii) (to the best of our knowledge) this is the first attempt to generate SEM-based fractional ventilation parameters for three different gases and two different under-sampling patterns (FGRE and X-Centric); (iv) no significant difference was found between the fractional ventilation estimates generated from the accelerated full-echo and half-echo imaging methods and, therefore, X-Centric can be safely used for the dynamic ventilation imaging of the short T_2_ * gases such as SF_6_. 

To our knowledge, this is the first demonstration of small animal lung fractional ventilation measurements generated using an alternative to the Deninger method. In this study, we demonstrated the feasibility of SEM-based accelerated ^3^He/^129^Xe/^19^F dynamic ventilation measurements with AF = 10 and 14 using examples of normal animals. A pixel-by-pixel comparison of Deninger’s approach and the SEM-derived fractional-ventilation-estimates obtained for AF = 10 and 14 (≤16% difference) has confirmed that even at AF = 14 the accuracy of the estimates is high enough to consider this method for prospective measurements. This is a promising result for the potential clinical translation of the ^19^F stretched-exponential model, which is ideally performed in a single breath-hold 3D isotropic voxel multi wash-out breath ^19^F MRI measurement. Note, retrospective under-sampling is certainly a limitation of this work, but it is not expected to be a limitation going forward to prospective studies in future; keeping in mind that the 3D k-space sampling will require sensitive RF coils [24,25,26,27,28,29,30] to ensure sufficient SNR of the 3D ^19^F lung images. 

The probability density function (Figure 3) was used to generate the SEM-based mean fractional-ventilation parameter. The shape of this function was consistent with the previously published probability density function obtained for the diffusivity distributions [17]. Unsurprisingly, the fractional-ventilation values obtained by the Deninger method based and SEM-based (0.22 ± 0.12 vs. 0.22 ± 0.08; *p* > 0.05; ^129^Xe, Table 2) were similar to the accelerated (X-Centric, AF = 14) case (0.21 ± 0.10 vs. 0.21 ± 0.09; *p* > 0.05; ^129^Xe, Table 2). The significant difference between the mean r values obtained with SEM and the Deninger method for ^3^He gas, is likely due to the smaller rat population compared with the ^129^Xe and ^19^F rat populations. The overall mean SEM *r* estimates generated for the ^19^F MRI lung data were reasonably similar to the previously reported estimates [18]. 

There are a number of limitations of this work. First of all, the ^3^He/^129^Xe MRI wash-in dynamic lung images were considered as the wash-out images and they did not normalize on the RF pulse “history” and oxygen-induced decay, thus the generated fractional ventilation estimates were lower than the previously reported [5,6]. We tried to mimic the ^19^F wash-out data using high quality ^3^He/^129^Xe dynamic ventilation images to understand how SNR affects the accuracy of the regional estimates (*r*) and that is why ^3^He/^129^Xe MRI wash-in dynamic lung images were not corrected. Unsurprisingly, the dynamic ^3^He/^129^Xe/^19^F images had a different SNR level, which affected the pixel-by-pixel difference showing the larger difference for the lower SNR images (Table 3). Nevertheless, the lowest SNR (5) for the highest number of wash-out breath was still sufficient to yield reasonable fractional ventilation values. However, we must admit that the dynamic ventilation images (^19^F study) for rat-5 had very low SNR and, as a result, it is likely the reason for the worst absolute mean difference between the *r* values generated from the original data sets and retrospectively the under-sampled/reconstructed datasets (Appendix A). Appendix A show that the high SNR data demonstrates the smallest absolute mean difference (<10% for ^3^He data and <16% for ^129^Xe data), while the ^19^F data had a wide distribution of the absolute mean differences. This is an important result showing a limitation of the proposed approach, specifically the SNR limitation. 

Furthermore, we used normal animals in this work, so the homogeneous distribution of the fractional ventilation estimates across the lungs was expected. This result is not specific to normal animals as a recent study of the rat models of inflammation and fibrosis disease [1] has suggested that the fractional ventilation maps obtained for sick animals can be homogeneous as well (Ref. [1], Figure 4). We have to admit that the lack of any diseased model of animals is a study limitation. Another limitation of this work is the lack of data analyses using a combination of Deninger’s equation with CS. Since there is no theoretical background for the CS Deninger approach presently, we could not use the prior knowledge (Deninger’s equation of signal behavior) to compensate for the significant under-sampling and to generate the probability functions similar to the CS SEM approach.

One more important question is over the influence of the significant k-space under-sampling on the image resolution. We would like to emphasize that the small number of the acquired k-space lines did not restrict us from sampling the high frequency line or the edge of k-space, moreover, the sparsity pattern varied from one image to another. The image reconstruction used all the acquired dynamic ventilation images and prior knowledge about the system through the SEM equation. This approach permitted us to ensure the nominal image resolution and, therefore, all the generated fractional ventilation maps had the expected nominal resolution as well. The small values of the mean absolute difference obtained for the ^3^He data set (AF = 10 and AF = 14) support this. The recent resolution phantom study has demonstrated that the significantly accelerated (AF = 10 and 14) dynamic ventilation measurement using the CS combined with SEM reconstruction did not lead to image resolution degradation [31]. It has also shown the benefit of signal averaging with the prior knowledge approach combined with the CS-based reconstruction. Basically, the significant acceleration, normally leading to the image resolution degradation when reconstructing each image independently, was compensated by a large number of wash-out images acquired with a varied sparsity pattern used for the group reconstruction powered by the prior knowledge SEM approach. 

Finally, Equation (3) was not obtained analytically, so an analytical solution may be possible in order to correlate the Deninger method-based and SEM-based fractional ventilation estimates.

Imaging strategies using parallel imaging [25,30], the phased receive arrays [29,30], and CS have permitted lung morphometry measurements that overcome the slower diffusion of xenon compared with helium and enabled whole lung ^129^Xe multi-b diffusion-weighted measurements in a single breath hold [14,32,33,34,35]. It has recently been shown that the combination of SEM with CS [21] (^129^Xe clinical study, one healthy subject) permitted under-sampling in both spatial and diffusion-sensitizing directions and was able to achieve imaging at AF = 10, while still providing accurate morphometry estimates [36]. Furthermore, the feasibility of SEM-based accelerated ^129^Xe morphometry with AF = 10 and 14 has been prospectively demonstrated in a small cohort of normal and irradiated rats [17].

In summary, SEM-based dynamic ventilation measurements can be significantly accelerated (up to 14x) without compromising the quality of generated biomarkers such as the fractional ventilation values. Both accelerated and unaccelerated dynamic ventilation ($r_{SEM}$) values using SEM with ^19^F MRI in normal rats agree well to previously published fractional ventilation estimates. This suggests that the SEM may be used as an alternative to the Deninger method in the case of normal animals and potentially for a number of other small animal lung disease models such as inflammation and fibrosis [37]. Finally, CS combined with the SEM permits a significant acceleration in the scan time for the ^3^He/^129^Xe/^19^F dynamic ventilation measurements and, therefore, should be considered for the characterization of lung function, especially in human subjects where breath hold durations may be limited due to the lung disease including the COVID-19 lung damage [38]. High quality ^3^He and ^129^Xe data suggest that the highly accelerated dynamic ventilation measurements still ensure the accurate fractional ventilation estimates.

## Figures and Tables

**Figure 1 diagnostics-13-00506-f001:**
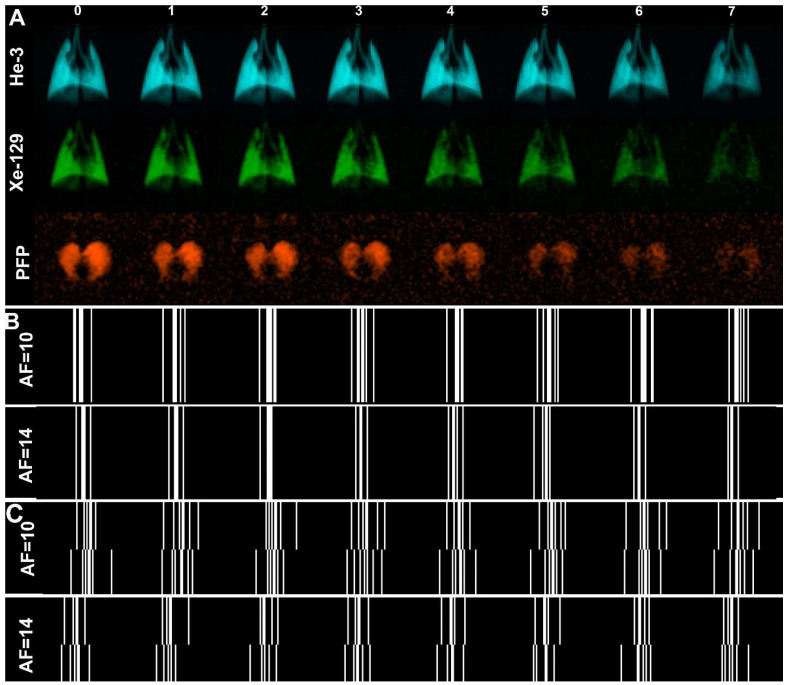
Wash-out ^3^He/^129^Xe/^19^F MRI Images obtained in Normal Rats. (**A**) depicts representative eight wash-out images obtained using ^3^He/^129^Xe/^19^F. K-space under-sampling in the phase encoding direction (fully sampled in the x-direction) schemes, ensuring a variety of sparsity patterns for each wash-out image is depicted by (**B**) AF = 10 and AF = 14 obtained for FGRE, and (**C**) AF = 10 and AF = 14 obtained for x-Centric that were retrospectively applied in wash-out direction. ^3^He/^129^Xe data were obtained using wash-in scheme. We reversed the image order to be able to fit the dynamic data with the SEM equation requiring the signal decay not signal growing. No under-sampling in the x-direction was used for FGRE while 50% under-sampling (compensated by doubling the lines in the phase encoding direction) was employed for x-Centric.

**Figure 2 diagnostics-13-00506-f002:**
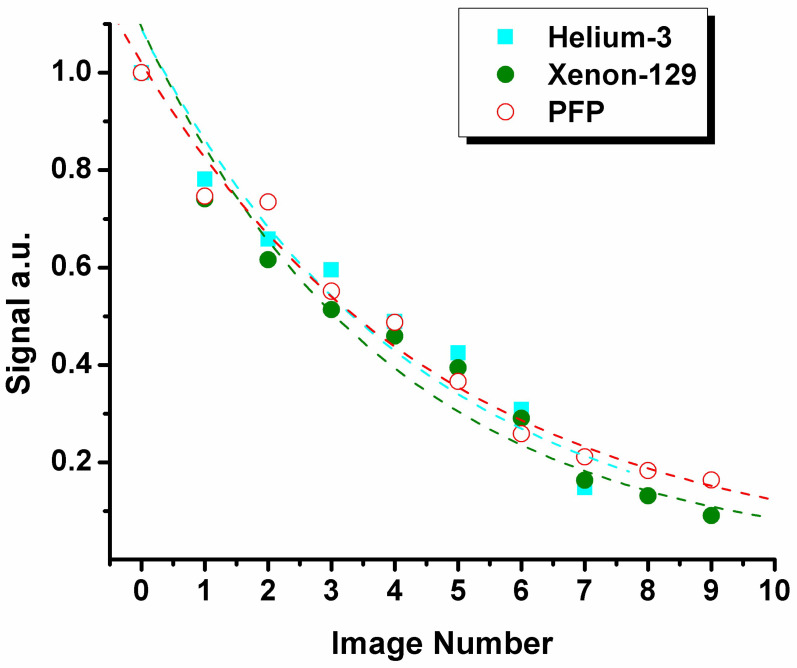
Bulk signal intensity dependence as a function of image number obtained from wash-out rat lung images (Figure 1A). The dashed lines show the best-fit of mono exponentials obtained from Figure 1A.

**Figure 3 diagnostics-13-00506-f003:**
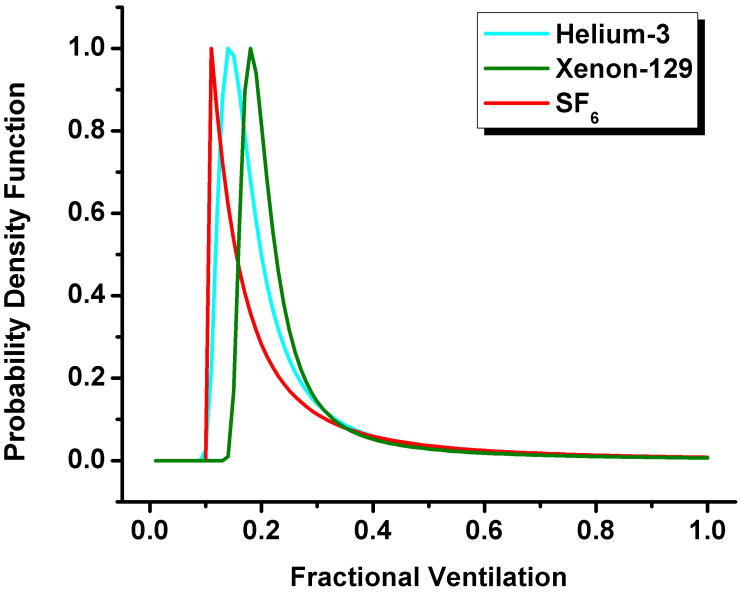
Normalized regional fractional-ventilation distributions obtained for representative animals ventilated with three different gases. Bulk fractional-ventilation distributions obtained for ^3^He (mean r′ = 0.22, mean β = 0.98, $r_{SEM}$ = 0.26, cyan line), ^129^Xe (mean r′ = 0.24, mean β = 0.90, $r_{SEM}$ = 0.28, dark green line), and ^19^F (mean r′ = 0.14, mean β = 0.76, $r_{SEM}$ = 0.24, red line) gases from three different animals. The plot shows the smallest $r_{SEM}$ peak value for ^19^F contract agent and largest $r_{SEM}$ peak value for ^129^Xe contract agent (0.11 vs. 0.18 for peak values, respectively). r′ = apparent MRI fractional-ventilation estimate; β = MRI-derived heterogeneity index; $r_{SEM}$ = MRI-derived SEM-based regional fractional-ventilation; SEM = stretched exponential model. The noticeable difference between β generated from the ^3^He/^129^Xe MRI and ^19^F MRI measurements is likely due to the use of two different ventilators and different ventilator settings are dictated by the physical property of the three gases.

**Figure 4 diagnostics-13-00506-f004:**
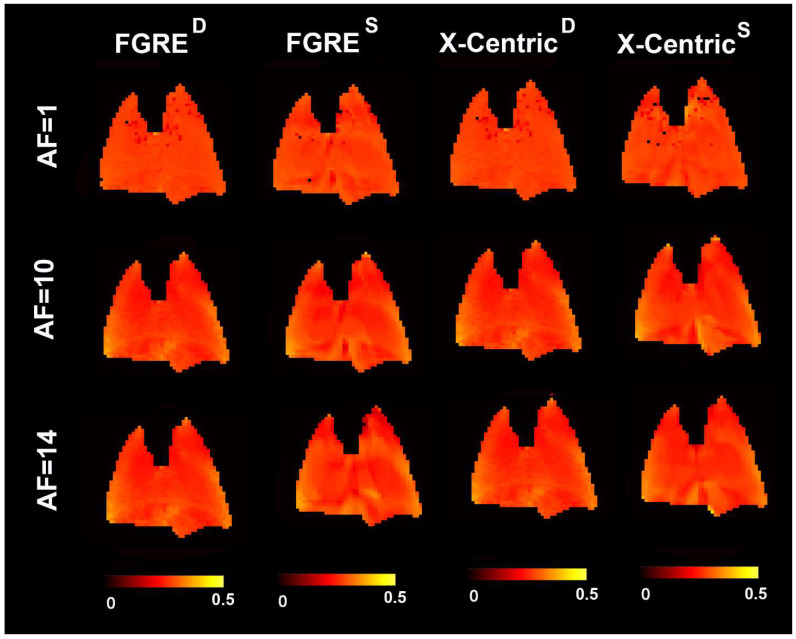
Representative ^3^He MRI-based fractional ventilation maps generated by using the Deninger method (^D^) and the SEM (^S^) from normal animal using two different imaging approaches (FGRE and X-Centric). The top panel shows fractional ventilation maps calculated for the original fully sampled k-space. The middle and bottom panels show the maps generated for the retrospectively under-sampled data mimicking AF = 10 and 14, correspondingly.

**Figure 5 diagnostics-13-00506-f005:**
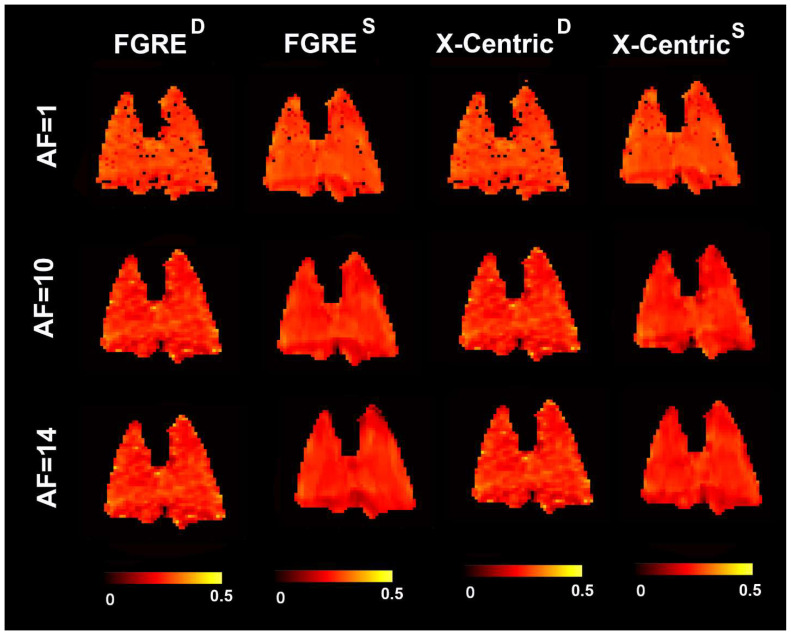
Representative ^129^Xe MRI-based fractional ventilation maps generated by using the Deninger method (^D^) and the SEM (^S^) from normal animal using two different imaging approaches (FGRE and X-Centric). The top panel shows fractional ventilation maps calculated for the original fully sampled k-space. The middle and bottom panels show the maps generated for the retrospectively under-sampled data mimicking AF = 10 and 14, correspondingly.

**Figure 6 diagnostics-13-00506-f006:**
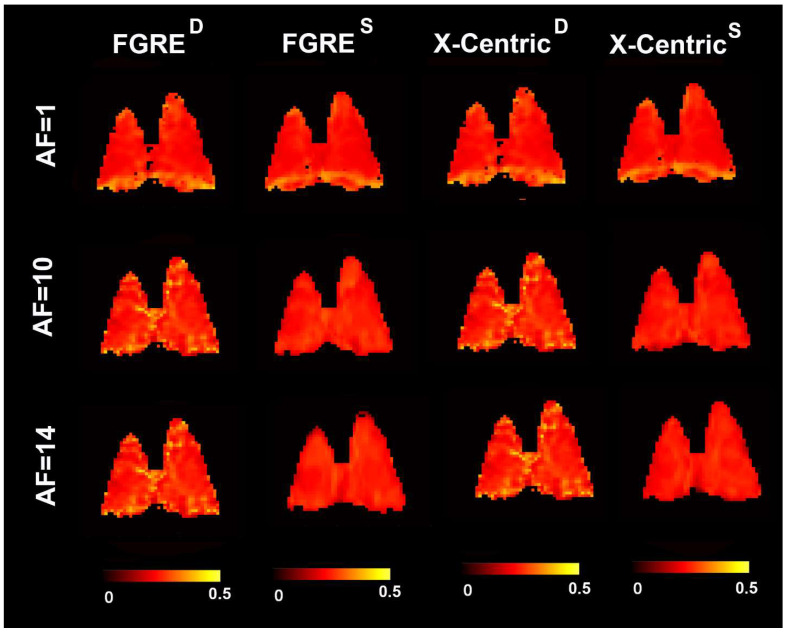
Representative ^19^F MRI-based fractional ventilation maps generated by using the Deninger method (^D^) and the SEM (^S^) from normal animal using two different imaging approaches (FGRE and X-Centric). The top panel shows fractional ventilation maps calculated for the original fully sampled k-space. The middle and bottom panels show the maps generated for the retrospectively under-sampled data mimicking AF = 10 and 14, correspondingly.

**Table 1 diagnostics-13-00506-t001:** ^3^He MRI-based Fractional-Ventilation Measurements (average from all rats).

	FGRE					X-Centric				
	r	r_sem_	r − r_sem_	r − r^A^	r_sem_ − r_sem_^A^	r	r_sem_	r − r_sem_	r − r^A^	r_sem_ − r_sem_^A^
AF = 1	0.22 (0.011)	0.20 (0.006)	5.0%	-	-	0.22 (0.011)	0.20 (0.006)	5.0%	-	-
AF = 10	0.22 (0.013)	0.20 (0.007)	6.5%	4.5%	4.0%	0.22 (0.013)	0.20 (0.008)	7.5%	4.5%	4.0%
AF = 14	0.22 (0.013)	0.20 (0.014)	7.0%	7.5%	6.0%	0.22 (0.011)	0.20 (0.007)	8.0%	5.0%	4.0%

r = MRI mean fractional ventilation estimate obtained with the Deninger method using fully sampled data; r_sem_ = MRI mean fractional ventilation estimate obtained with SEM using fully sampled data; r^A^ = MRI mean fractional ventilation estimate obtained with the Deninger method using accelerated data; r_sem_^A^ = MRI mean fractional ventilation estimate obtained with SEM using accelerated data; r **−** r^A^/r **−** r_sem_/r_sem_
**−** r_sem_^A^/ = pixel-by-pixel deference between the fractional ventilation maps; SEM = stretched exponential model; AF = acceleration factor.

**Table 2 diagnostics-13-00506-t002:** ^129^Xe MRI-based Fractional-Ventilation Measurements (average from all rats).

	FGRE					X-Centric				
	r	r_sem_	r − r_sem_	r − r^A^	r_sem_ − r_sem_^A^	r	r_sem_	r − r_sem_	r − r^A^	r_sem_ − r_sem_^A^
AF = 1	0.22 (0.01)	0.22 (0.01)	6.5%	-	-	0.22 (0.01)	0.22 (0.01)	6.4%	-	-
AF = 10	0.22 (0.01)	0.22 (0.01)	12.0%	7.0%	4.5%	0.21 (0.01)	0.22 (0.01)	13%	7.5%	6.0%
AF = 14	0.21 (0.01)	0.22 (0.01)	13.0%	10.0%	7.0%	0.21 (0.01)	0.21 (0.01)	13%	8.0%	5.0%

r = MRI mean fractional ventilation estimate obtained with the Deninger method using fully sampled data; r_sem_ = MRI mean fractional ventilation estimate obtained with SEM using fully sampled data; r^A^ = MRI mean fractional ventilation estimate obtained with the Deninger method using accelerated data; r_sem_^A^ = MRI mean fractional ventilation estimate obtained with SEM using accelerated data; r **−** r^A^/r **−** r_sem_/r_sem_
**−** r_sem_^A^/ = pixel-by-pixel deference between the fractional ventilation maps; SEM = stretched exponential model; AF = acceleration factor.

**Table 3 diagnostics-13-00506-t003:** ^19^F MRI-based Fractional-Ventilation Measurements (average from all rats).

	FGRE					X-Centric				
	r	r_sem_	r − r_sem_	r − r^A^	r_sem_ − r_sem_^A^	r	r_sem_	r − r_sem_	r − r^A^	r_sem_ − r_sem_^A^
AF = 1	0.24 (0.02)	0.24 (0.04)	15%	-	-	0.24 (0.02)	0.24 (0.04)	15%	-	-
AF = 10	0.22 (0.02)	0.22 (0.04)	16%	12.5%	8%	0.22 (0.013)	0.20 (0.008)	12.5%	14.0%	9%
AF = 14	0.21 (0.02)	0.22 (0.04)	10%	14%	12%	0.22 (0.011)	0.20 (0.007)	14.0%	15.0%	9%

r = MRI mean fractional ventilation estimate obtained with the Deninger method using fully sampled data; r_sem_ = MRI mean fractional ventilation estimate obtained with SEM using fully sampled data; r^A^ = MRI mean fractional ventilation estimate obtained with the Deninger method using accelerated data; r_sem_^A^ = MRI mean fractional ventilation estimate obtained with SEM using accelerated data; r **−** r^A^/r **−** r_sem_/r_sem_
**−** r_sem_^A^/ = pixel-by-pixel deference between the fractional ventilation maps; SEM = stretched exponential model; AF = acceleration factor.

## Data Availability

Data are not available due to the ethical restrictions.

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
