# Peer review of "Feasibility of Dynamic Inhaled Gas MRI-Based Measurements Using Acceleration Combined with the Stretched Exponential Model"

_diagnostics, 2023, doi:10.3390/diagnostics13030506_

Round 1

Reviewer 1 Report

The article is very interesting and well written.

I think it should be accepted for the publication.

Reviewer 2 Report

In this manuscript Sembhi and colleague have explored the feasibility of using a stretched exponential model (SEM) approach in combination with compressed sensing (CS) acceleration in deriving fraction ventilation measurements from dynamic ventilation MRI. Retrospective undersampling of 3He/129Xe/19F dynamic ventilation data acquired in normal rats was performed and mean fractional-ventilation values from fully-sampled and undersampled data were compared. Furthermore, the SEM approach to fitting dynamic ventilation data was benchmarked against the established Deninger’s approach for fully-sampled and undersampled data through a pixel-by-pixel comparison. A relatively small difference between the two approaches was observed, and it was concluded that the SEM approach even at AF=14 was accurate enough for prospective acquisitions.

The implementation of the SEM fit to derive fractional ventilation is an interesting novel approach as the SEM has predominantly been applied to diffusion-weighted MR imaging across different organ systems with proton and hyperpolarized gas MRI. However, I feel in the manuscript the motivation behind why this approach is considered is missing or not well-described. In addition, I feel a prospective demonstration of the CS undersampling in conjunction with the SEM fitting approach is required to fully conclude that the AF=14 CS with the SEM approach is feasible for dynamic ventilation MRI with inhaled gases. Further details around these main points and some other comments are described below.

Introduction:

- Overall, the introduction is a little disjointed and I would recommend re-wording or rearranging to flow better.

- 19F dynamic lung imaging is mentioned in the 1st paragraph but then fluorinated gases are introduced as an alternative to HP gases in the 2nd paragraph.

- There needs to be more discussion of the motivation behind using the SEM approach and CS undersampling for dynamic inhaled gas MRI in the introduction. The current introduction doesn’t provide much context into why the feasibility of this approach was investigated and why is it different to the Deninger approach

Theory:

- The beta value (heterogeneity index) is not reported in the subsequent results tables. Presumably this value is a marker for the intra-voxel fractional ventilation heterogeneity – analogous to intra-voxel diffusion heterogeneity when the SEM is used in diffusion MRI?

- In figure 3 for three representative rats the beta value varies quite a bit between inhaled gas. This would suggest differences in ventilation heterogeneity in normal rats? Do the authors have any suggestions on the reason behind this? Did the authors notice a systematic difference in beta across all rats for the different gases?

- What is the reason behind the analytical derivation of the probability distribution of fractional ventilation p(r)? It appears only r’ and beta from eq 3 are reported in the results. The p(r) from eq 4-6 is not reported or used to derive further metrics in the manuscript and only briefly mentioned in the discussion. Have the authors explored or considered using the expectation value from the probability distribution?

Methods:

- The number of rats used for each type of inhaled gas needs to be stated. The abstract mentions 13 rats in total but from the supplementary tables I can see 5 (3He), 7 (129Xe), 6 (19F). Please also indicate which rats were imaged with more than one type of inhaled gas.

- As mentioned earlier, what is the motivation behind accelerating dynamic ventilation MRI? My understanding of the techniques is that it requires wash out of the inhaled gas over 9 breath-holds. If an acquisition was accelerated the total time of the scanning wouldn’t change as the same number of breath holds are required? Are the authors planning to acquire higher resolution prospective data with time saved from CS undersampling?

- Did the authors test other retrospective undersampling patterns (i.e. ones between AF=2 to 10)? Would testing a larger range of acceleration factors be more beneficial in determining the undersampling effect on the fractional ventilation measurements?

Results:

- Why was the SEM approach used to fit the undersampled data? Is there an advantage of the SEM approach over the Deninger’s one? From results it can be seen that the ‘ground truth’ Deninger’s fractional ventilation estimate can be derived from both fully-sampled and under-sampled data already. Furthermore there is a small mean absolute difference between Deninger’s and SEM approaches, and this should be discussed in the Discussion section.

- Could the colour scale of the ventilation maps be adjusted so subtle differences between undersampling and fitting methods be observed

- Please clarify in the text or caption that the Tables 1,2 3 are the average from all rats imaged with the respective inhaled gas. Currently the table captions are the same as the supplementary ones that have individual rat results

- Please check the results text is referring the correct table. I can see Table 1S (3He) is being referenced in the 129Xe results section

- Minor comment - Table 1S seems to be in the main manuscript rather than supplementary material - could just be an issue on my end when downloading the manuscript and supplementary material documents?

Discussion:

- While the authors do acknowledge that prospective CS acquisition is a limitation of the manuscript, I feel prospective acquisition of dynamic ventilation MRI data is required to truly demonstrate the CS and SEM approach is feasible for fractional ventilation measurements. Different flip angles and sequence timings etc, as well as SNR considerations will be required for a prospective undersampled acquisitions that are not accounted for in retrospective undersampling. I feel this needs to be demonstrated to conclude that high acceleration (AF=14) is feasible in fractional ventilation measurements from dynamic ventilation MRI.

Reviewer 3 Report

The authors describe a study in which they implement an alternative method for modeling (SEM) gas-phase wash-out imaging. This enabled inclusion of prior knowledge into a compressed sensing reconstruction. They implemented this reconstruction for multiple different gases, acquisition methods, and undersampling patterns. The images were then fit to the same model to obtain fractional ventilation maps and were compared to fully sampled images modeled with the standard Deninger method.

Overall, the science is interesting and worthy of publication. However, there is a need for a lot of improvements to be ready for publication. From the non-scientific point of view, there are grammar issues throughout the manuscript. Additionally, the flow and consistency when switching between the He/Xe and F gases could use a lot of work. From the scientific standpoint, some of the decisions on how to assess the model could be done be better. Additionally, some important information to truly understand the method is missing.

Broad Comments:

1.         There are many areas of incorrect grammar throughout the paper. See some examples in specific comments 2, 10, and 12 for examples.

2.         There are a decent number of mistakes which I am assuming are typos. See some examples in specific comments 11, 13, and 26 for examples.

3.         The flow could be improved. The paper jumps around a lot and in some places, F is discussed first, and in others, He/Xe is discussed first. And in some places, it isn’t clear which one is being discussed. That makes it difficult to follow.

4.         At the end of the manuscript, it states that the Xe and He images were used to assess the model and its behavior with respect to SNR. If that’s the case, I would strongly suggest swapping out the Xe/He data for simulations with different SNRs. That would permit comparisons to a known value, the ability to test different wash-out rates, different SNRs, inhomogeneous distributions, etc. Additionally, this would allow the Xe/He portions to be removed, avoiding the confusion mentioned above.

5.         A number of the important references are to conference abstracts. Ideally, these would be replaced with peer-reviewed publications which are easily accessible and provide the full information necessary to understand the specifics.

Specific Comments:

1.         Page 1: “13 normal-rats”; Were these the same 13 rats for all measurements? Doesn’t appear to be based on the tables.

2.         Page 1: “were studied … were performed”; Incorrect grammar/run-on sentence.

3.         Page 1: “This wash-out scheme”; This free-breathing scheme?

4.         Page 2: “informative (trachea and bronchi)”: How are those informative? And wasn’t the entire lungs imaged as mentioned later in the sentence?

5.         Page 2: “these methods provide”: I thought this was in reference to the SSFP imaging but the rest of the sentence doesn’t match or make sense.

6.         Figure 1: “Wash-out”: Much later in the paper, we are informed these He/Xe images are wash-in images that were then time-reversed. This is misleading.

7.         Figure 1: B/C: Why does the AF 14 sampling pattern have two patterns? This isn’t discussed at all. Also, where are kmax/kmin for these patterns? That become important later in the discussion.

8.         Page 4: “SEM approach providing the extra parameter”: How was this implemented? References aren’t specific enough to provide the necessary information. I’m assuming this was implemented like the SIDER method. How was the model initialized, i.e., what were the initial assumptions/guesses for B, C, r`, and beta?

9.         Figure 3: Normalizing these PDFs, rather than a.u., and a comparison to the true histogram is ideal to observe how well the model does would be very beneficial.

10.       Figure 5: “19F animal”: The animal isn’t 19F, the gas inhaled by the animal was 19F.

11.       Page 6: “were repeated three times”: Later in the paper contradicts this.

12.       Page 6: “inert fluorinated gas/oxygen”: This is repeated, and the sentence does not make grammatical sense.

13.       Page 6: “50G/cm, slew rate =2000T/m/s”: The use of varying units here and later as well as not using the industry standard units is not ideal. Additionally, check the conversions as the values for the scanners don’t seem correct.

14.       Page 6: “before the transfer … and vacuumed”: This likely is not what you mean; this sentence indicates vacuuming after transferring the He.

15.       Page 7: “3He imaging”: T2 decay doesn’t cause diffusion-induced signal attenuation. Also, was only 3He completed with a short echo time? Not sure what this is attempting to inform the reader of.

16.       Page 7: “after delivering … after image”: Poor grammar. Use “and” or remove the after image portion.

17.       Page 7: “chosen to based on”: Poor grammar and second half of sentence is incomplete.

18.       Eq: 7: Why was absolute difference used? I would think this would penalize the low SNR images unfairly. Consider adding a measure without the absolute?

19.       Page 8: “Multivariate analysis of variance”: What are the multivariables? Only once is listed. Also, as the undersampled and fully sampled images are paired, why wasn a repeated measure ANOVA or similar not used? Also P<0.05 is mentioned throughout, but is that just the result of the MANOVA? Or were ad-hoc tests used and, if so, which ones?

20.       Page 11: “mean absolute differences…”: I found the reporting of the mean absolute difference confusing. Considering rearranging/simplifying the text throughout the manuscript.

21.       Tables 1-3: Some parts of the tables do not make sense to me. For example, how can r (column 1), the fully sampled Denninger method, vary for AFs? I think another variable is necessary to indicate Denninger method, independent of sampling.

22.       Table 1S. Should be moved to SI with 2S and 3S.

23.       Page 12: “ventilation values obtained all rats”: This does not make grammatical sense.

24.       Page 13: “Table 2). Table 1S”: I believe Table 1 and 3He is a copy issue and weren’t updated to 2S and 129Xe. This happens multiple times. Table 2S is discussed, unnecessarily(?), later.

25.       Page 15: “very sensitive RF coils”: Not sure what this brings to the discussion as that’s a requirement for 2D as well. Also 22-26 are all He and Xe coils – not 19F.

26.       Page 15: “smallest mean … >10% ... >16%”: Should be <?

27.       Page 15: “small number of the acquired”: It does not restrict you from sampling high frequencies and is still completed in the read-out dimension. However, if the sampling patterns are accurate, it does not appear that any high-frequencies in the phase-encoding direction is performed.

28.       Page 15: “all generated fractional maps”: The same nominal resolution was never investigated or proven. This is also where simulations could be useful

Round 2

Reviewer 2 Report

I would like to thank Sembhi and colleagues for their prompt revision of this manuscript. Most of my previous points/concerns have been addressed. However, there are still a few points/concerns that still remain and are detailed below:

Introduction

- Minor comment. Page 2: “This prior knowledge permits to increase the acceleration factor…” needs grammatical re-wording

Methods

- Thank you for the clarification regarding the derivation of the mean fractional ventilation estimate. I would suggest the authors also state in the image processing and analysis methods section that this value is calculated as the expectation value of P(r).

- As a follow-up to Reviewer 3 comment 6 regarding the clarification of wash-in data for He/Xe and wash-out data for 19F. I would suggest this clarification is also stated in the methods section as well as the Figure 1 caption.

- I would like some more clarification on the k-space sampling patterns used in this work. The methods state two k-space masks were used to retrospectively undersampling k-space data (AF=10, AF=14). However, in Figure 1B AF=10 is for FGRE sampling and Figure 1C AF=14 is for x-Centric. Were there another two k-space sampling patterns for AF=10 x-Centric and AF=14 FGRE? If so, they should be shown in the manuscript as well.

- The He/Xe and 19F datasets have different number of images (8 vs 12), were the 1st 8 k-space patterns the same for each gas but the last 4 different? If so, these additional k-space patterns should also be shown

- Is the AF=10 FGRE k-space pattern identical to that in Figure 1 of Ref 17? If so, this should be clearly stated in the manuscript.

- It’s not clear in the methods which rats had SF6 or PFP gases. Did all 6 rats who underwent 19F MRI have both? Are the fractional ventilation results for 19F MRI (Figure 6, Table 3) a mixture of SF6 and PFP?

- Minor comment – “Sprague-Dawley rats were used for this study and were prepared using methods described previously [6, 18]” I would suggest re-wording this sentence as it suggests that new rats were used in this study and prepared as described previously when the rats used are actually those from ref 6 and 18.

Results

- “We thank the reviewer for the comment, unfortunately we are not very clear how to adjust the scale as the fractional ventilation varies between zero and one.”

As the average fractional ventilation value is ~0.25, perhaps the authors could display the fractional ventilation maps between 0 and 0.5 to better visualise the variation of values across the lungs.

Discussion

- “We agree with reviewer that the Deninger’s equation predicting the signal behaviour can be potentially combined with CS permitting the large acceleration factors. However, this is required deriving a number of the analytical equation prior to utilizing which is out of scope of this work.”

“Finally, Eqn. (3) was not obtained analytically, so an analytical solution may be possible in order to correlate the Deninger method based and SEM-based fractional ventilation estimates.”

I don’t think I understand this response and the sentence in the discussion. The fractional ventilation maps in Figure 4, 5 & 6 show retrospectively undersampled Deninger methods maps for both FGRE and x-Centric sampling. This would suggest that analytical equations have been used already for CS combined with Deninger? This would appear to be within the scope of the work and should be discussed in the manuscript. Perhaps if there is some difference in computational time or complexity between SEM and Deninger methods that can be discussed?

- Minor comment – “...that retrospective under-sampling is certainly a limitation of this work, but it is not expected to be a limitation going forward to prospective studies in future; keeping in mind that the 3D k-space sampling will require sensitive RF coils [24-30] to ensure sufficient SNR of the 3D 19F lung images”. This sentence would suggest that future prospective acquisitions will be with 3D images. If so, I would recommend the authors re-perform retrospective simulations with 3D k-space undersampling patterns.

- Minor comment – “ Furthermore, the feasibility of SEM-based accelerated 129Xe morphometry with AF=10 and 14 has been prospectively demonstrated in a small cohort of normal and irradiated rats [32].” The wrong reference appears to be cited here - Ref 32 refers to AF=7 undersampling in human subjects.

Reviewer 3 Report

The authors have made significant improvements to the manuscript following suggestions and comments from the reviewers. However, there appears to be a few miscommunications and comments which were not addressed from Reviewers 2 and 3. Those are discussed below.

1.     While the grammar in the manuscript has been significantly improved, it appears the added text suffers from the same issue as the original manuscript.

2.     Some responses stated the topic is out of scope for the work and didn’t address it at all. Even if investigating something is out of scope for the work, there are often still good reasons to discuss them. For example, even if not combining Deninger’s equation with CS, discussing why SEM was chosen over the conventional Deninger equation is important. Was it due to the complexity of the equation or another reason? What is the benefit of SEM over Deninger CS?

3.     Even if fractional ventilation varies from 0 to 1, that does not need to be the range of color scale in the figures. It could be ~0.15-0.3 to show the variations more clearly. Similarly, it is clear that each dynamic has a different sampling pattern, but Figure 1 B and C should have the kmax and kmin shown along the x-axis for each dynamic in order for the reader to visualize how close each phase encode is to kmax/min. It appears that, with the sampling patterns here, the phase encodings only reach ~50% of kmax which would lead to blurring in the phase encoding direction.

4.     With the lack of any prospective demonstration or application to data with ventilation abnormalities, the results should be well supported. Suggested improvements to the supporting data was largely not implemented. Suggestions included analysis of different AFs, comparing measured to calculated probability densities, use of a digital phantom simulation to permit comparison to a known true fractional ventilation map (ideally with structure for assessment of resolution), additional image metrics, etc. These all would provide additional support and confirmation for the application of the method.

5.     Compressed sensing is known to suffer from blurring, albeit much less than a lower resolution image acquisition for the same scan time. With these higher AFs, there is likely non-negligible blurring. Thus, claiming same nominal resolution will need to be supported. Absolute difference assesses the whole image and indicates accurate fractional ventilation values are obtained. Edge blurring would only result in small changes as the edges are sparse and is taken advantage of in many CS applications. SSIM, image differences, and even simple line profiles are commonly used in CS applications to assess edges.

6.     “The fully sample Denninger method corresponds to AF1...” This does not clear up the tables for me. There are 4 different r’s listed in each table; r, rsem, rA, and rsemA. r is defined as “MRI mean fractional ventilation estimate obtained with the Deninger method using fully-sampled data”. Thus, to my understanding, there should only be one value for r for FGRE and one for X-centric. The r’s for AF=10 and 14 do not make sense as they are not fully-sampled. Are the r values listed under r actually rA for AF=10 and 14?

7.     The statistics are still vague. The additional variables taken into consideration for the MANOVA are not clear and the statistics should have been done as paired/repeated which is not clear. A quick estimated t-test for r vs rsem for AF = 1 in He shows a very significant difference (P < 0.01). That’s before taking into consideration the acceleration and CS reconstructions. Also, AF = 1 isn’t in the SI tables making this calculation not possible.

Round 3

Reviewer 3 Report

The authors have adequately modified the paper for publication.
